# Enhanced RNA replication and pathogenesis in recent SARS-CoV-2 variants harboring the L260F mutation in NSP6

Taha Y. Taha[1,2☉], Shahrzad Ezzatpour[3,4☉], Jennifer M. Hayashi[1☉], Chengjin Ye[5☉], Francisco J. Zapatero-Belinchón[1], Julia A. Rosecrans[1], Gabriella R. Kimmerly[1], Irene P. Chen[1,10], Keith Walcott[1], Anna Kurianowicz[1], Danielle M. Jorgens[6], Natalie R. Chaplin[6], Annette Choi[4], David W. Buchholz[4], Julie Sahler[4], Zachary T. Hilt[4], Brian Imbiakha[4], Cecilia Vagi-Szmola[1], Mauricio Montano[1], Erica Stevenson[1,7,8,9], Martin Gordon[1,7,8,9], Danielle L. Swaney[1,7,8,9], Nevan J. Krogan[1,2,7,8,9], Gary R. Whittaker[4], Luis Martinez-Sobrido[5]*, Hector C. Aguilar[3,4]*, Melanie Ott[1,10,11]*

1 Gladstone Institutes, San Francisco, California, United States of America, 2 Department of Bioengineering and Therapeutic Sciences, University of California, San Francisco, California, United States of America, 3 Department of Microbiology, College of Agriculture and Life Sciences, Cornell University, Ithaca, New York, United States of America, 4 Department of Microbiology and Immunology, Cornell University College of Veterinary Medicine, Ithaca, New York, United States of America, 5 Texas Biomedical Research Institute, San Antonio, Texas, United States of America, 6 Electron Microscope Laboratory, University of California, Berkeley, California, United States of America, 7 Department of Cellular and Molecular Pharmacology, University of California, San Francisco, California, United States of America, 8 Quantitative Biosciences Institute (QBI), University of California, San Francisco, California, United States of America, 9 Quantitative Biosciences Institute (QBI) COVID-19 Research Group (QCRG), San Francisco, California, United States of America, 10 Department of Medicine, University of California, San Francisco, California, United States of America, 11 Chan Zuckerberg Biohub – San Francisco, San Francisco, California, United States of America

☉ These authors contributed equally to this work.
* lmartinez@txbiomed.org (LMS); ha363@cornell.edu (HCA); melanie.ott@gladstone.ucsf.edu (MO)

## Abstract

The COVID-19 pandemic has been driven by SARS-CoV-2 variants with enhanced transmission and immune escape. Apart from extensive evolution in the Spike protein, non-Spike mutations are accumulating across the entire viral genome and their functional impact is not well understood. To address the contribution of these mutations, we reconstructed genomes of recent Omicron variants with disabled Spike expression (replicons) to systematically compare their RNA replication capabilities independently from Spike. We also used a single reference replicon and complemented it with various Omicron variant Spike proteins to quantify viral entry capabilities in single-round infection assays. Viral entry and RNA replication were negatively correlated, suggesting that as variants evolve reduced entry functions under growing immune pressure on Spike, RNA replication increases as a compensatory mechanism. We identified multiple mutations across the viral genome that enhanced viral RNA replication. NSP6 emerged as a hotspot with a distinct L260F mutation independently arising in the BQ.1.1 and XBB.1.16 variants. Using mutant and revertant NSP6 viral clones, the L260F mutation was validated to enhance viral replication in cells and increase pathogenesis in mice. Notably, this mutation reduced host lipid droplet content by NSP6. Collectively, a systematic analysis of RNA replication of recent

**Data availability statement:** All relevant data are within the manuscript and its Supporting Information files.

**Funding:** This work was supported by the National Institutes of Health (U19AI135990 and U19AI135972 to N.J.K and grant F31AI164671-01 to I.P.C.). We gratefully acknowledge support from the Roddenberry Foundation, P. and E. Taft, Schmidt Futures, Gordon and Betty Moore Foundation, and the James B. Pendleton Charitable Trust to M.O.. M.O. is a Chan Zuckerberg Biohub – San Francisco Investigator. The funders did not play a role in the study design, data collection and analysis, decision to publish, or preparation of the manuscript.

**Competing interests:** I have read the journal's policy and the authors of this manuscript have the following competing interests: TYT and MO are inventors on a patent application filed by the Gladstone Institutes that covers the use of pGLUE to generate SARS-CoV-2 infectious clones and replicons. MO is a cofounder of DirectBio and on the SAB for Invisishield. The NJK Laboratory has received research support from Vir Biotechnology, F. Hoffmann-La Roche, and Rezo Therapeutics. NJK has a financially compensated consulting agreement with Maze Therapeutics. NJK is the President and is on the Board of Directors of Rezo Therapeutics, and he is a shareholder in Tenaya Therapeutics, Maze Therapeutics, Rezo Therapeutics, GEn1E Lifesciences, and Interline Therapeutics. All other authors declare no competing interests.

Omicron variants defined NSP6's key role in viral RNA replication that provides insight into evolutionary trajectories of recent variants with possible therapeutic implications.

## Author summary

As SARS-CoV-2 continues to spread and adapt in humans, viral variants with enhanced spread and immune evasion have emerged throughout the COVID-19 pandemic. While most of the mutations occur in the Spike protein and have been extensively studied, non-Spike mutations have been accumulating and are not as well understood. Here, we constructed Spike-defective genomes of recent Omicron variants and systematically compared their RNA replication capabilities independently from Spike. We also performed single-round infection assays with various Omicron variant Spike proteins to quantify viral entry capabilities. Interestingly, viral entry and RNA replication were negatively correlated, suggesting that as variants evolved reduced entry functions under growing immune pressure on Spike, RNA replication increased as a compensatory mechanism. We focused on a viral protein NSP6 that acquired mutations that significantly enhanced RNA replication. We validated that a frequently accessed L260F mutation in NSP6 enhanced viral infection in cells and increased pathogenesis in mice. The mutation enhances NSP6's role in hijacking the host lipid droplet machinery for viral replication. Collectively, we highlight the important role of non-Spike mutations in the evolutionary trajectories of SARS-CoV-2 variants with possible monitoring and therapeutic implications.

## Introduction

The COVID-19 pandemic has been driven by SARS-CoV-2 variants with enhanced spread and immune escape [1,2]. Over the past 3 years, Omicron variants have continuously evolved their Spike proteins towards enhanced immune escape at the cost of reduced entry, with the most recent JN.1-derived variants having over 60 nonsynonymous mutations in Spike [3,4]. Evolution outside of Spike has also been significant with over 40 nonsynonymous mutations [5], but the impact of these mutations on viral replication is less well understood as compared to those arising in Spike. Since the emergence of the BA.2 variant, many mutations across the genome persisted, and subsequent variants acquired additional mutations, suggesting continued positive selection (Fig 1A and 1B). It is difficult to fully ascertain the impact of these mutations in the context of fully replicating viruses due to the dominant and significantly different Spike phenotypes across variants [6]. Determining the specific impact of these mutations independently of Spike is necessary to better understand the evolutionary trajectory of SARS-CoV-2.

We previously developed a Spike replicon system to investigate the impact of mutations across the genome independently of Spike [7–10]. We and others have reported that Omicron BA.1 mutations in NSP6 (LSG 105-107 deletion (ΔLSG) and I189V) attenuated viral replication due to the reduction of lipid droplet (LD) content in infected cells [7,11]. Similar to other RNA viruses, coronaviruses hijack the host's LD machinery to enhance viral replication via NSP6's interaction with the host protein DFCP1 [12]. Over the past two years, these detrimental BA.1 mutations in NSP6 were quickly replaced with a SGF 106-108 deletion (ΔSGF) that is persistent in all Omicron variants post-BA.2 (Fig 1A). Notably, this deletion emerged previously in other SARS-CoV-2 variants but not in Delta, Epsilon, Mu, and BA.1

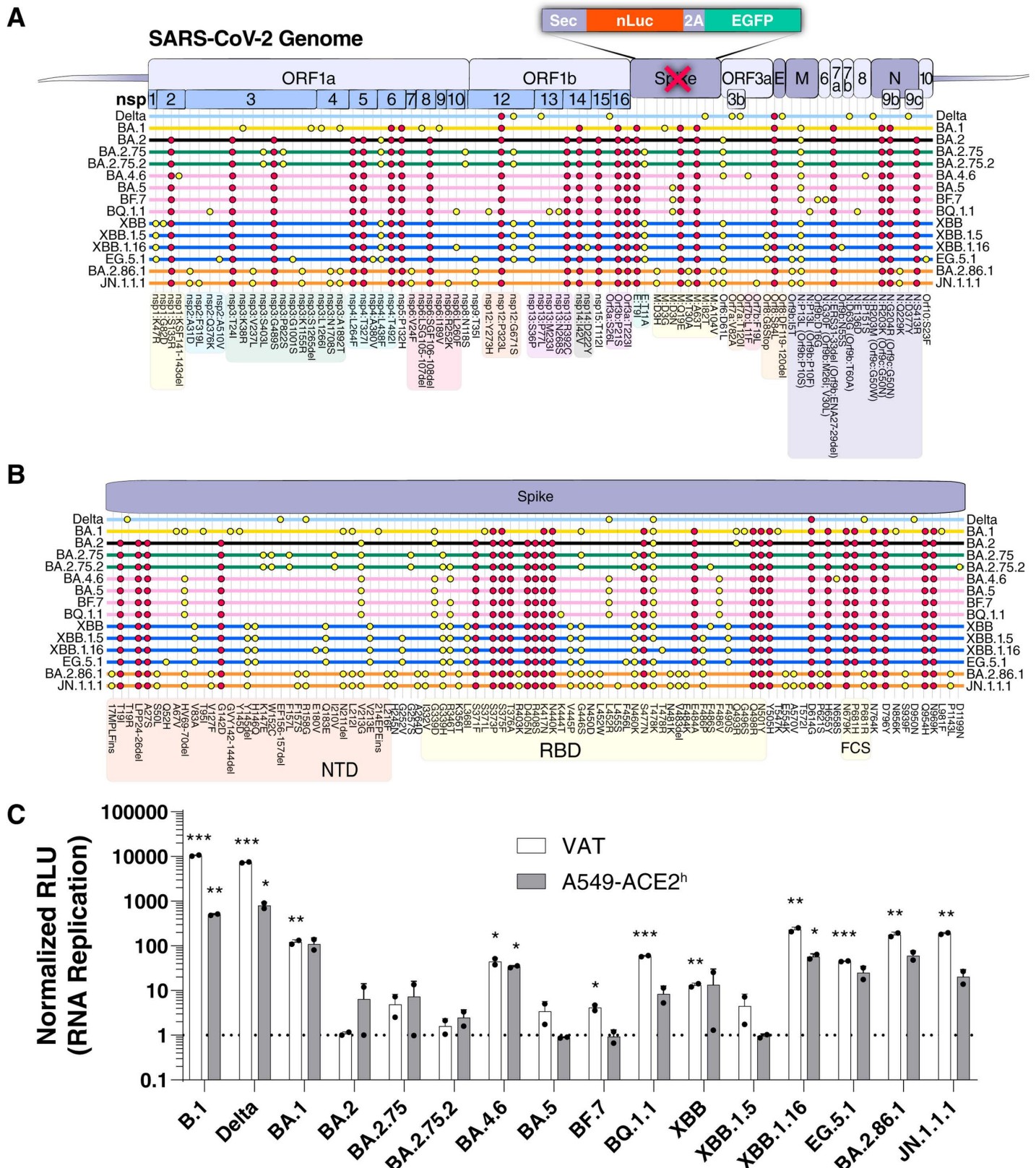

**Fig 1. Recent Omicron variants have higher replication than early ones. A and B)** Consensus non-Spike (A) and Spike (B) mutations (>95% of sequences at time of emergence) in all dominant SARS-CoV-2 variants post-Delta. BA.2 fixed mutations are indicated in red. **C)** RNA replication measurement of each variant paired

with its concordant Spike expression vector using SARS-CoV-2 Spike replicons. The data is plotted as mean +/- SD of two independent biological replicates conducted in triplicate. *, p<0.05; **, p<0.01; ***, p<0.001; ****, p<0.0001 by two-sided Student's T-test. NTD: N-terminal domain; RBD: receptor binding domain; FCS: furin cleavage site; VAT: Vero cells stably expressing ACE2 and TMPRSS2; A549-ACE2h: A549 cells stably expressing high levels of ACE2.

[13]. BA.2-derived variants independently acquired additional mutations in NSP6 (Fig 1A), suggesting that ΔSGF alone might not be optimal for replication within the Omicron genetic background. In one case, BA.5 and its descendant BQ.1.1, as well as the independent variant XBB.1.16, converged on the L260F mutation in NSP6 indicating a potential fitness advantage for this particular mutation.

To define the evolution in RNA replication of recent dominant Omicron variants, we performed single-round infections with a comprehensive set of replicons and Spike proteins to dedifferentiate between RNA replication and viral entry capacities. We demonstrate that variants before BA.2.86 progressively enhanced RNA replication in the context of lowered Spike entry functions. The L260F mutation in NSP6 contributed to enhanced viral RNA replication and LD content reduction. The mutation also significantly enhanced variant pathogenesis as reverting it in the BA.5 or XBB.1.16 variants decreased viral replication and enhanced survival *in vivo*. Together, our data identify NSP6 as a central evolutionary rheostat for fine-tuned RNA replication in response to reduced Spike entry functions.

## Results

### SARS-CoV-2 Omicron variants pre-BA.2.86 evolved towards enhanced RNA replication and reduced Spike entry

To determine the impact of mutations across the genome on viral replication for dominant SARS-CoV-2 Omicron variants, we constructed Spike replicons and corresponding Spike expression vectors and measured the combined replication phenotype for each variant in single-round infection assays (Fig 1A–C). All the Omicron variants replicated less than B.1 and Delta variants (Fig 1C). Within the Omicron variants, BA.2-derived (BA.2, BA.2.75, BA.2.75.2) and BA.5-derived (BA.5, BF.7) variants had the lowest replication, while BA.1, BA.4.6, BQ.1.1, XBB, and EG.5.1 replicated to an intermediate level, and XBB.1.16, BA.2.86.1, and JN.1.1.1 replicated to the highest level within the Omicron variants (Fig 1C). Interestingly, the variants within the BA.5 and XBB lineages emerged with initially low replication capacities but gained additional mutations that improved their replication phenotypes in the subsequent subvariants BQ.1.1 and XBB.1.16, respectively (Fig 1C). Notably, both BQ.1.1 and XBB.1.16 variants converged on an NSP6 L260F mutation without a similar convergence in Spike mutations (Fig 1A and 1B), suggesting that this mutation confers a fitness advantage.

To delineate the role of Spike vs non-Spike mutations, we measured entry and RNA replication of variants separately by either combining different Spike replicons with a fixed Spike protein or a fixed replicon with different Spike proteins (Fig 2A). We utilized Vero cells stably expressing ACE2 and TMPRSS2 (VAT cells) that support high levels of entry/replication or A549 cells stably expressing high levels of ACE2 only (A549-ACE2h cells) with overall lower levels of entry/replication. As expected, pre-Omicron variants B.1 and Delta exhibited high levels of entry and replication in both cell types (Fig 2B and 2C). The slightly lowered levels of entry for the Delta variant in A549-ACE2h cells reflects the TMPRSS2-dependence of this variant [14,15]. Omicron variants showed significantly lower entry and replication levels, except for BA.2.86-derived variants (BA.2.86.1 and JN.1.1.1), which exhibited high entry and replication rates in both cell types (Fig 2B and 2C) consistent with previous reports [16,17]. Notably, several Omicron variants entered A549-ACE2h cells more efficiently than

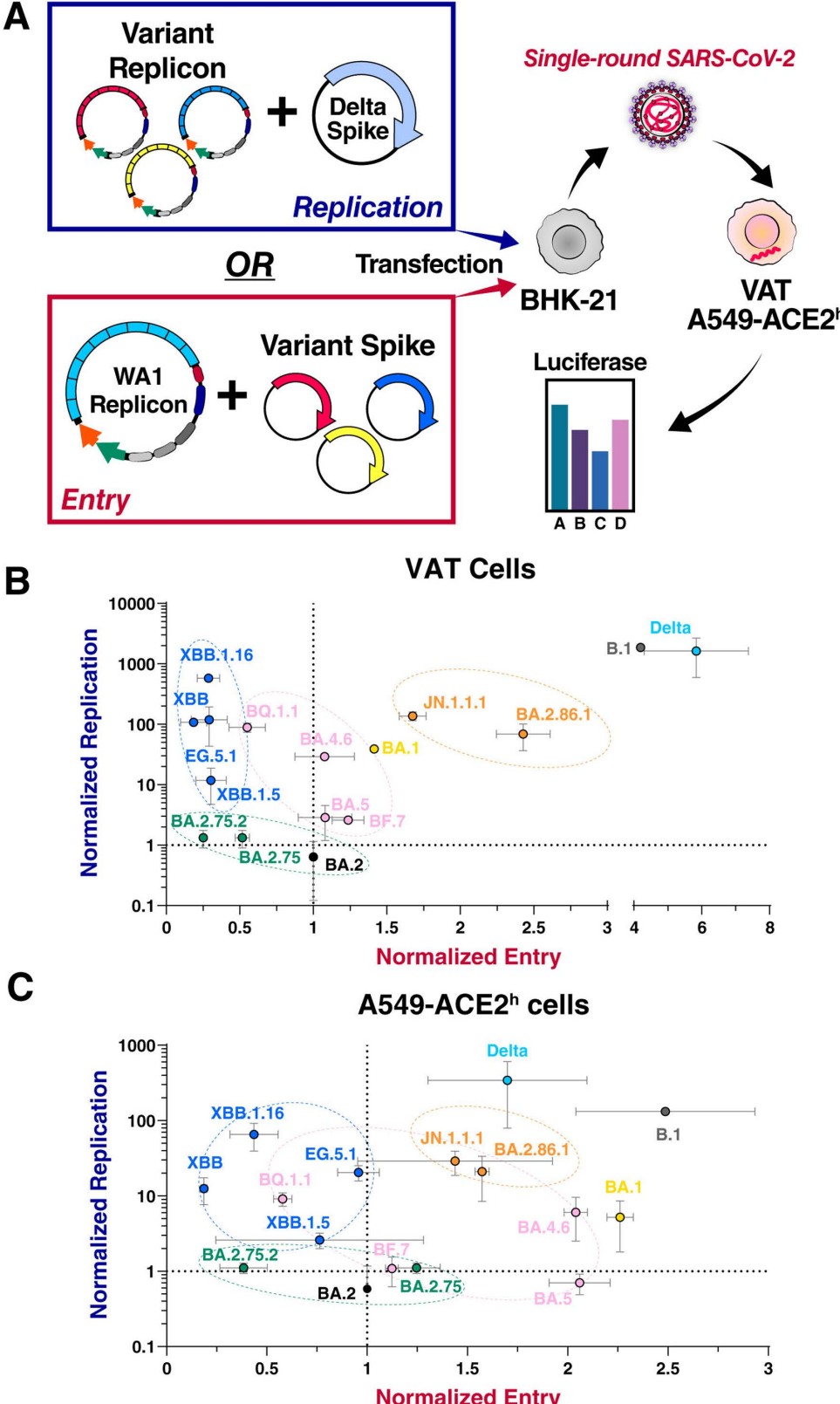

**Fig 2. RNA replication and entry negatively correlate for Omicron variants. A)** Schematic of replicon experiments to measure RNA replication and entry. **B and C)** Normalized RNA replication and normalized entry for all variants

were measured in VAT (B) and A549-ACE2$^h$ (C) cells and graphed on a scatter plot. Data is plotted as mean +/- SD of two independent biological replicates each conducted in triplicate. VAT: Vero cells stably expressing ACE2 and TMPRSS2; A549-ACE2$^h$: A549 cells stably expressing high levels of ACE2.

VAT cells (Fig 2B and 2C), which reflects that the endosomal route is the preferred mode of entry for these variants [14,15]. Importantly, entry and RNA replication correlated negatively for BA.2-, BA.5-, XBB-, and BA.2.86-derived subvariants (Fig 2B and 2C). For the BA.5- and XBB-derived variants, the highest replicating variants were BQ.1.1 and XBB.1.16, respectively (Fig 2B and 2C), both showing the L260F mutation in NSP6 (Fig 1A). Another NSP6 mutation (R252K) occurred in the JN.1.1.1 variant, a derivative of BA.2.86.1 with lower entry and higher RNA replication capacities than its ancestral variant (Fig 2B and 2C); this variant also contains additional mutations (NSP2:F319L, Orf7b:F19L). Currently circulating JN.1-derived KP.3.1.1 and XEC variants have not gained additional NSP6 mutations but appear to have further enhanced their Spike entry and immune evasion functions [18]. Collectively, these data show evolutionary trajectories towards enhanced RNA replication in recent Omicron variants with NSP6 being a possible mutational hotspot.

## NSP6 mutations enhance viral RNA replication and lipid content reduction

Next, we tested the impact of NSP6 mutations on different Omicron replicons (Fig 3A). Addition of the NSP6 L260F onto the BA.2 variant replicon resulted in ~1000-fold enhancement in RNA replication to almost similar levels as XBB.1.16 (Fig 3B). Correspondingly, reversion of the mutation in the XBB.1.16 variant replicon reduced replication ~100-fold to above BA.2 replication levels (Fig 3B). This confirms that the NSP6 L260F mutation is sufficient to enhance RNA replication within the XBB.1.16 variant. However, other mutations within XBB-derived variants may contribute to enhanced RNA replication independently of NSP6. We therefore tested additional mutations found in XBB-derived variants (NSP1:K47R, NSP1:G82D, NSP12:G671S, NSP13:S36P) both in BA.2 and XBB variant backbones. We found that all mutations enhanced replication in both backgrounds and that NSP1 mutations had the strongest additive phenotype (S1 Fig). NSP1 is essential for preferential translational inhibition of host vs. viral transcripts [19] and therefore could enhance viral replication by translation of higher levels of viral proteins. These data are consistent with a previous report showing high replication levels mediated by the NSP12 G671S mutation in the Delta variant [20]. Reversion of characteristic NSP6 mutations in BA.2.86.1 and JN.1.1.1 did not significantly alter replication for the V24F mutation but reduced replication by 3-fold for the R252K mutation, respectively (Fig 3B). Collectively, these data demonstrate a positive albeit variable role of NSP6 mutations on RNA replication that may have evolved to counterbalance negative evolutionary pressures on other proteins such as Spike in recent Omicron variants.

NSP6 has previously been implicated in reducing LD content in infected cells [12], and we have shown that mutations in BA.1 dampen this function [7]. To test the effect of the L260F mutation on LD properties, we measured intensities of LD staining, number, and size in cells infected with replicons by high-throughput microscopy (Figs 3C and S2). As expected, BA.1 infected cells showed higher levels of LD staining than the earlier B.1 or the XBB.1.16 variants, indicating higher LD content (Figs 3C and S2). Interestingly, BA.2-infected cells had even higher numbers of LD and staining levels than those infected with BA.1, suggesting that the NSP6 ΔSGF deletion, present in BA.2, further decreases the LD reduction function of NSP6 (Figs 3C and S2). Addition of the NSP6 L260F mutation onto the BA.2 replicon background reduced LD staining in infected cells to B.1 and XBB.1.16 variants' levels (Figs 3C and S2). Conversely, reversion of the mutation in the XBB.1.16 replicon increased LD staining in

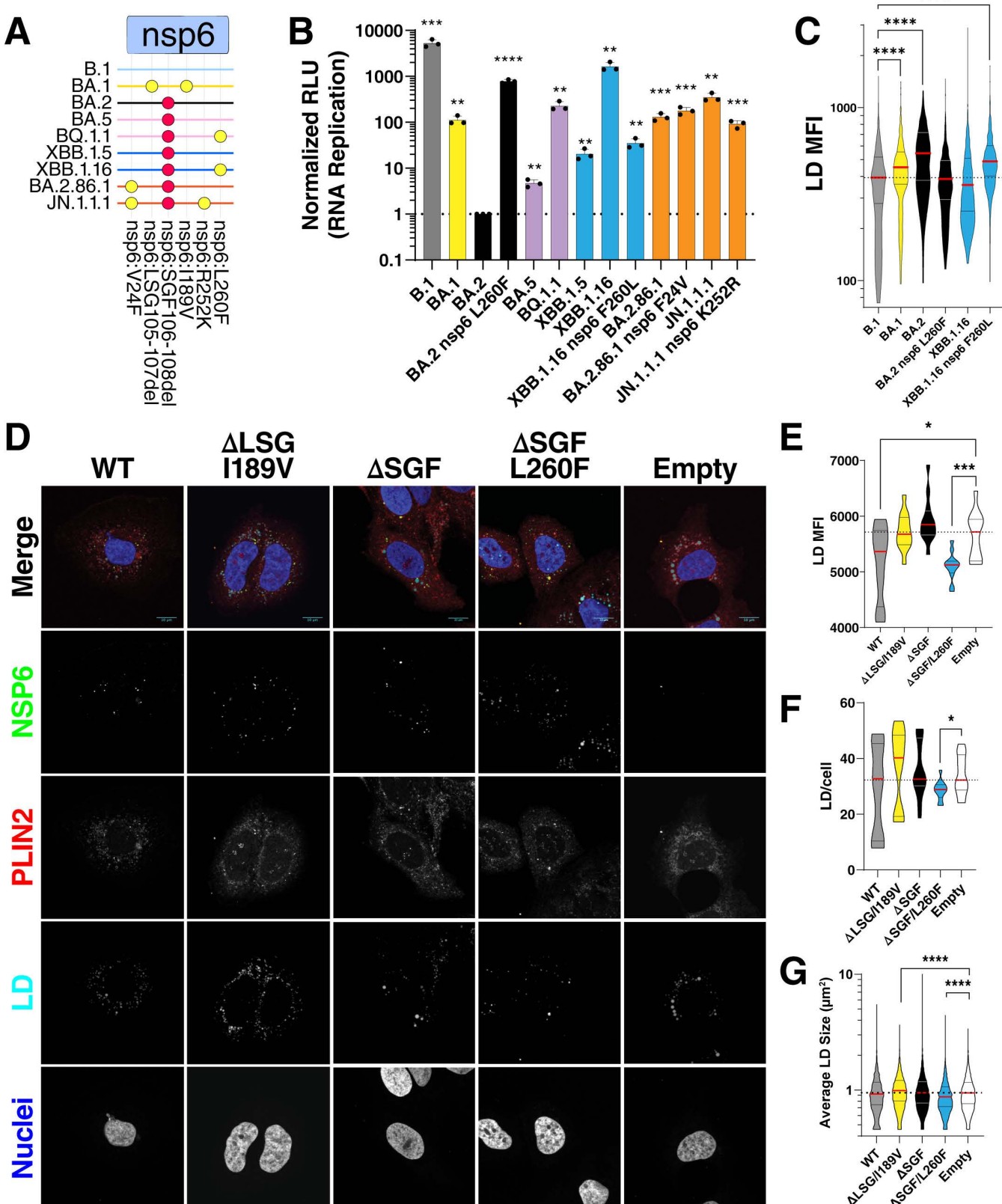

**Fig 3. NSP6 L260F mutation enhances RNA replication and reduces LD content in infected cells. A)** NSP6 mutations in Omicron variants. BA.2 fixed mutations are indicated in red. **B)** RNA replication measurement of indicated variants in VAT cells. Data is plotted as mean +/- SD of three independent

biological replicates each conducted in duplicate. **C)** Quantification of the relative LD mean fluorescent intensity (MFI) per dsRNA positive cells in images shown in S2 Fig using box and whiskers plot, and comparisons were made as indicated by two-sided Student's T-test. **D)** Immunofluorescence staining of HuH-7.5 cells stably expressing indicated NSP6 proteins and stained for LD, PLIN2, and NSP6. **E, F, and G)** Quantification of the relative LD MFI, LD number, and average LD size per NSP6-positive cells in images shown in D using box and whiskers plot, and comparisons were made as indicated by two-sided Student's T-test. For panels using box and whiskers plots, interquartile range (IQR) of boxplot is between 25th and 75th percentiles and center line indicates median value. Whiskers of boxplot is extended to the maxima and minima. Maxima is the largest value and minima is the smallest value in the dataset. LD, lipid droplet. *, p<0.05; **, p<0.01; ***, p<0.001; ****, p<0.0001 by two-sided Student's T-test.

infected cells to similar levels as found for the BA.2 variant (Figs 3C and S2). These data demonstrate that variants containing NSP6 L260F significantly reduce LD content corresponding to enhanced RNA replication.

To validate that the NSP6 L260F mutation is sufficient for reduced LD content independent of other mutations across the genomes, we constructed NSP6 expression vectors bearing the BA.1 (ΔLSG/I189V), BA.2 (ΔSGF), and XBB.1.16 (ΔSGF/L260F) mutations. We generated HuH-7.5 cell lines stably expressing these mutants and conducted immunofluorescence staining for NSP6, fatty acids and PLIN2 as a LD marker. All NSP6 mutants were similarly expressed and localized in a characteristic punctate pattern (Fig 3D). With high-throughput quantitative imaging, we found that, compared to empty vector-expressing cells, LD staining was significantly decreased in NSP6 wild-type and ΔSGF/L260F expressing cells but not in the NSP6 ΔLSG/I189V and ΔSGF expressing cells (Fig 3E). Notably, both the number of LDs per cell and the average LD size were significantly lower in cells expressing NSP6 ΔSGF/L260F, supporting the critical role of this mutation in modulating LD dynamics (Fig 3F and 3G). Because the NSP6 L260F mutation is located in the C-terminus of the protein outside the ER membrane towards the cytoplasm, we tested interaction of the NSP6 proteins with the previously reported LD biogenesis protein DFCP1 [12,21]. All NSP6 proteins interacted with this protein and we did not detect any differences between the wildtype and mutant proteins (S3 Fig). Collectively, these data support the model that the positive role of the NSP6 L260F mutation in viral RNA replication is coupled with reduced LD content by NSP6.

## NSP6 L260F is critical for viral infection *in vitro* and pathogenesis *in vivo*

Throughout our studies, we utilized replicons to rapidly and safely study the role of non-Spike mutations on viral RNA replication. To determine whether our findings translate to replication-competent SARS-CoV-2, we constructed an XBB.1.16 virus naturally containing the NSP6 L260F mutation and a revertant virus (XBB.1.16 NSP6 F260L) predicted to have lower replication than the parental virus (Fig 3B). Inspection of the plaque morphology of infected VAT cells demonstrated that the revertant virus has a significantly reduced plaque size compared with the parental XBB.1.16 virus (Fig 4A). In comparative infection experiments in A549-ACE2^h cells, XBB.1.16 NSP6 F260L showed 10- and 100-fold lower viral particle production and intracellular viral RNA at 24 and 48 hours post-infection as measured by plaque assay and RT-qPCR, respectively (Fig 4B). This confirmed the phenotype of the replicons in fully infectious viral clones. To validate the impact of NSP6 L260F *in vivo*, we switched to the BA.5 variant, which we previously reported a patient isolated BA.5 (hCoV-19/USA/COR-22-063113/2022) to be pathogenic in mice and also contains the NSP6 L260F mutation [22]. We constructed a revertant virus (BA.5 NSP6 F260L) and measured growth kinetics in VAT cells. Similar to the results obtained with the XBB.1.16 variant, BA.5 NSP6 F260L decreased viral particle production by 10-fold at 48 and 72 hours post-infection compared with the parental BA.5 variant (Fig 4C).

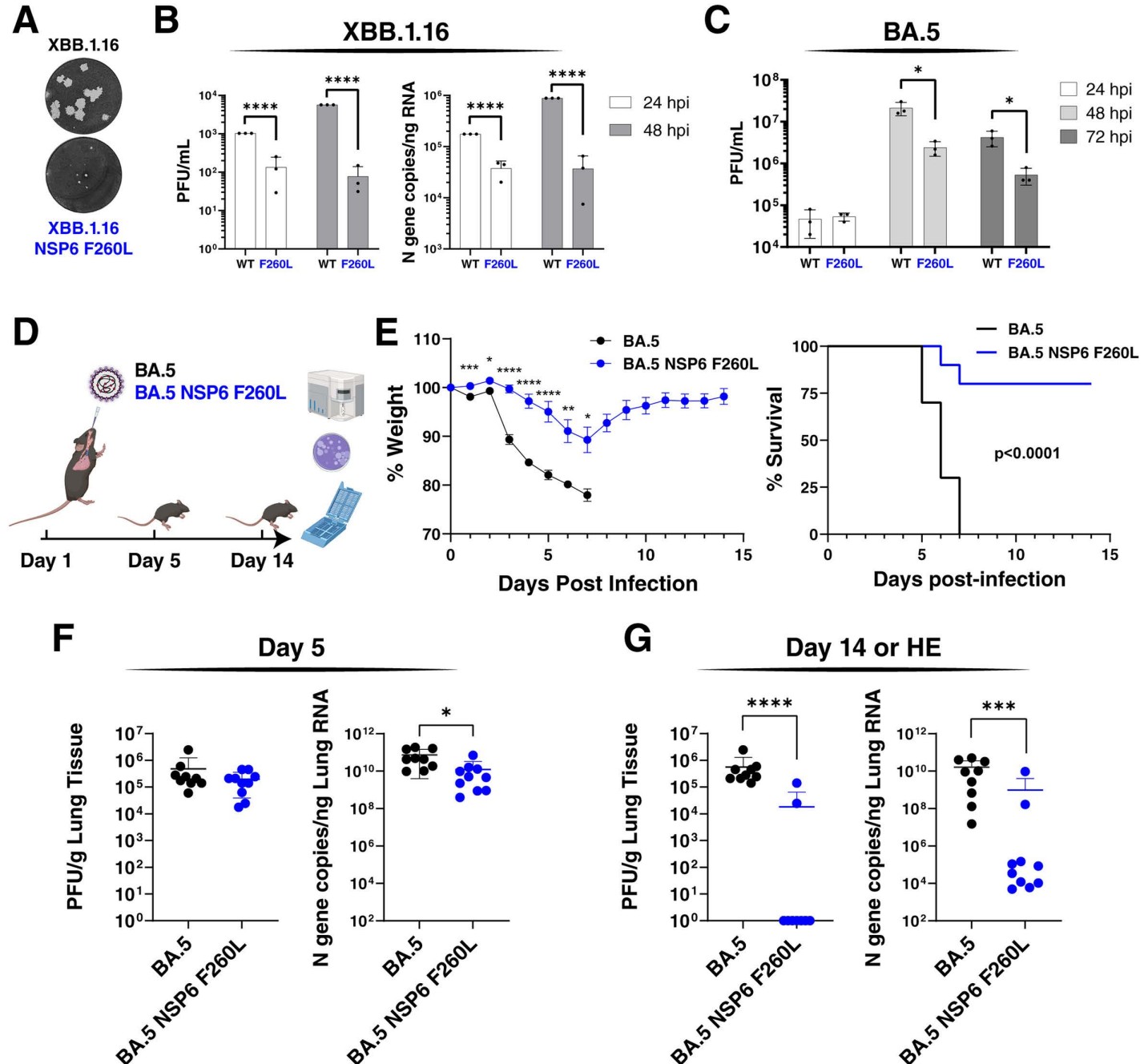

**Fig 4. NSP6 L260F is critical for viral infection *in vitro* and pathogenesis *in vivo*. A)** Plaque morphology of indicated viruses in VAT cells at 48 hours post-infection. The images were pseudocolored to black and white for optimal visualization. **B)** Particle production and intracellular viral RNA abundance were measured by plaque assay and RT-qPCR, respectively, at 24 and 48 hours post-infection in A549-ACE2[h] cells for indicated viruses. Data is plotted as mean +/- SD for three independent biological replicates each conducted in duplicate. **C)** Particle production was measured by plaque assay at 24, 48, and 72 hours post-infection in VAT cells for indicated viruses. Data is plotted as mean +/- SD of one representative biological replicate conducted in triplicate. **D)** Schematic representation of the *in vivo* experiment to measure the replication and pathogenesis of BA.5 and BA.5 NSP6 F260L viruses ($10^5$ PFU) in K18-hACE2 mice. n=10 per virus group per time-point. Clip art was created with BioRender.com with permission. **E)** Animal weight is plotted over the study time course as mean +/- SEM. BA.5 group animals all succumbed to infection at 7 days post-infection and therefore weight data are only available until that time-point. Survival was determined based on the humane endpoint (HE) of 20% weight loss and is plotted as a Kaplan-Meier curve. **F and G)** Particle production and intracellular viral RNA abundance were measured by plaque assay and RT-qPCR, respectively, at 5 (F) and 14 (G) days post-infection. For mice not reaching the end of the study, their tissues were collected when they reached the HE. Data is plotted as mean +/- SD. *, $p<0.05$; **, $p<0.01$; ***, $p<0.001$; ****, $p<0.0001$ by two-sided Student's T-test.

Next, we infected K18-hACE2 mice, overexpressing the human ACE2 receptor, with BA.5 and revertant NSP6 F260L viruses via intranasal application of $10^5$ infectious particles (Fig 4D). This dose has previously shown to be lethal in mice infected with BA.5 [22]. Over the study time course, the BA.5-infected mice demonstrated significant weight loss, and all mice succumbed to infection based on the humane endpoint of ≥20% weight loss at seven days post infection (Fig 4E). In contrast, mice infected with the BA.5 NSP6 F260L virus exhibited only a ~10% weight loss on average, and 80% of the mice survived by the end of the study (Fig 4E). This demonstrated a significant impact of the NSP6 L260F mutation on viral pathogenesis.

To determine whether survival was linked to reduced viral replication, we measured the abundance of viral particles and RNA in lung tissues at 5 and 14 days post-infection (or humane endpoint for BA.5-infected mice). Notably, at five days post-infection, the two viruses showed similar levels of viral particle production and only slightly lower levels of viral RNA for the BA.5 NSP6 F260L virus (Fig 4F). Flow cytometry analysis of all mice at this time point showed higher levels of immune cells in the BA.5 group compared with the BA.5 NSP6 F260L group in lymph node and spleen tissues (S4 Fig). Only at day 14 or at humane endpoint, BA.5 virus-infected tissues showed significantly higher levels of viral particles and RNA as compared to BA.5 NSP6 F260L-infected animals (Fig 4G). These data demonstrate that BA.5 and BA.5 NSP6 F260L infections *in vivo* show striking differences in pathogenesis with notable differences in viral replication at late time points after infection. Collectively, these findings underscore the critical role of the NSP6 L260F mutation *in vivo*.

## Discussion

Our study may provide explanations for evolutionary trajectories of Omicron variants. When we uncoupled variant RNA replication and entry capacities through replicon technologies, we found both negatively correlated in pre-BA.2.86 Omicron variants, but not pre-Omicron variants. We further investigated a single mutation in NSP6, a small transmembrane protein involved in ER zippering, DMV homeostasis, and lipid reduction. We found multiple lineages converged independently on L260F, which we show enhances RNA replication and lipid reduction. The mutation further confers heightened pathogenesis in mice, a phenotype not solely explained by the replication advantage and requires further studies *in vivo*.

SARS-CoV-2 evolution balances many factors impacting its fitness including entry, immune evasion, packaging, RNA replication, and others. It has been difficult to independently investigate these effects due to the dominant phenotype of Spike on replication [6] and the laborious task of constructing multiple combinations of Spike recombinant viruses across variants. To address this, we developed a replicon assay to study the impact of non-Spike mutations on viral RNA replication independently of Spike. This technology should prove useful to examine any mutation in nonstructural proteins in the context or out of context with its respective Spike proteins.

We also used the technology to systematically examine entry capabilities of Omicron variants by testing their respective Spike proteins. Notably, Omicron Spike proteins evolve under enormous immune pressure due to infections and vaccination [1,6]. To withstand this pressure, post-BA.2 variants evolved Spike proteins with reduced entry as shown here with single round replicon infections, and as reported by others [6], but enhanced immune evasion [1,23]. Interestingly, BA.2.86-derived variants contain an additional Spike glycosylation site (N354) that is predicted to enhance viral fitness due to multiple mechanisms regulating RBD conformational state, cell-cell fusion, and decreased immunogenicity [3]. Consistent with our data, subsequent lineages such as JN.1.1.1 have reduced entry due to additional RBD mutations that optimize immune evasion [17].

The NSP6 L260F mutation enhances viral replication in multiple variant backgrounds and has been observed in other variants throughout the pandemic ([24]). However, prior to Omicron variants, the mutation was never dominant suggesting potential epistasis within the Omicron genetic background. Indeed, NSP6 L260F addition to the BA.2 replicon rescued replication. We predicted the structure of all NSP6 variants using AlphaFold3 server ([25]) and found that ΔSGF mutation shifts the position of the second alpha helix on the luminal side of the ER (S5A and S5B Fig). However, the two mutations enhancing replication R252K and L260F are in the C-terminal cytosolic tail of NSP6 (S5A and S5C Fig) that potentially interacts with other host or viral proteins, but we did not detect differences in interactions with the known partner DFCP1 [12]. These observations suggest that NSP6 L260F may be accessible to current and future variants and can be used to tune replication in response to immune pressures. Interestingly, a similar mutation within the C-terminal domain of MERS NSP6 (L232F) was found to be associated with host jump to humans and enhances RNA replication without impacting DMV morphology or ER zippering [26]. Therefore, NSP6 mutations may play an important role in coronavirus evolution beyond SARS-CoV-2.

NSP6 is involved in ER zippering and filtering proteins and lipids to the DMVs where replication occurs [12]. However, the precise mechanisms of action remain unknown. We have previously reported that NSP6 has a lipid reduction function and that BA.1 mutations decreasing this function dampen replication [7]. Consistent with this model, the BA.1 mutations disappeared suggesting purifying selection and were replaced by the ΔSGF mutation in all post-BA.2 variants. However, many variants acquired additional NSP6 mutations suggesting that ΔSGF is not optimal within the Omicron variant background but potentially optimal for other variants pre-Delta [13]. Indeed, BA.2 has low RNA replication and additional NSP6 mutations in BQ.1.1, XBB.1.16, and JN.1.1.1 rectify this defect and have much higher replication than their ancestral strains. The most dominant mutation converged on by BQ.1.1 and XBB.1.16 is NSP6 L260F, which we found to enhance viral replication and reduce LD content in cells. A limitation of our experimental setup is that we cannot attribute the LD content reduction in infected cells or cells expressing NSP6 to altered LD biogenesis vs. enhanced LD consumption.

We detected a significant pathogenesis effect of NSP6 L260F as a revertant virus was significantly attenuated *in vivo* even though replication early post-infection was only slightly reduced. The exact role of NSP6 in pathogenesis remains unknown, but mutations in NSP6 BA.1 have previously been reported to decrease pathogenesis [11]. It is possible that NSP6 mutations enhance replication and this can lead to enhanced pathogenesis. This is supported by our finding that the revertant shows a replication defect at late stages of infection, but leaves replication kinetics at early stages unexplained. Another possibility is that NSP6 mutations affect pathogenesis independently from viral replication. This is supported by us finding more immune cells in spleen and lymph nodes of BA.5-infected animals despite equal replication dynamics. This could also explain the enhanced pathogenesis endowed by this mutation and the quick clearance of the revenant virus. A type I interferon antagonism function of NSP6 was previously reported [27] and is enhanced by the ΔSGF mutation. We show that this mutation has reduced viral replication and higher LD content in infected cells. Therefore, it is possible that different NSP6 mutations confer enhanced viral fitness by balancing the different functions of NSP6. Interestingly, in a recent study of American mink experimentally infected with SARS-CoV-2, the NSP6 L260F mutation was enriched in lung tissue over oral swab samples [28]. Therefore, the mutation could facilitate for the Omicron variant to adapt to the lower respiratory tract, which would be more pathogenic but likely lead to a decrease in transmission. This may explain why the L260F mutation did not get fixed in subsequent viral lineages, such as JN.1.1.1 and beyond, that may potentially be optimized for immune evasion

and transmission but not RNA replication and pathogenesis. Future studies will focus on dissecting the mechanism of NSP6 in viral replication, LD content dynamics, and pathogenesis.

## Materials and methods

### Ethics statement

All research conducted in this study complies with all relevant ethical regulations. All experiments conducted with viruses were performed in a certified biosafety level 3 (BSL3) laboratory and experiments were approved by the Institutional Biosafety Committee of the University of California, San Francisco and the Gladstone Institutes. All protocols concerning animal use were approved by Cornell University's Institutional Animal Care and Use Committee (mouse protocol no. 2017-0108 and BSL-3 Institutional Biosafety Committee no. MUA-16371-1) and conducted in strict accordance with the National Institutes of Health Guide for the Care and Use of Laboratory Animals.

### Sequence analysis

Viral sequences were downloaded from the GISAID database and analyzed for mutations and used to generate consensus sequences utilizing the Geneious Prime software version 2022.2.1.

### Plasmids, replicons, and infectious clones construction and rescue

All variants were constructed using pGLUE [7], except for BA.5 and BA.5 NSP6 F260L infectious clones which were constructed as described previously [29]. The BA.5 sequence was originally obtained through BEI Resources, NIAID, NIH: SARS-Related Coronavirus 2, Isolate hCoV-19/USA/COR-22-063113/2022 (Lineage BA.5; Omicron Variant), NR-58616, contributed by Dr. Richard J. Webby. The mCherry-DFCP1 plasmid was a gift from Dr. Do-Hyung Kim (Addgene plasmid #86746; http://n2t.net/addgene:86746; RRID:Addgene_86746).

### Replicon entry and replication measurements

All replicon experiments were conducted as described previously [7,8,30] with minor modifications. Briefly, $2x10^5$ BHK-21 cells were plated in 6-well plates. After 24 hours, the cells were transfected with 4 µg of replicon plasmid, 2 µg each Spike and Nucleocapsid R203M mutant expression vectors [31,32] using X-tremeGENE 9 DNA Transfection Reagent at 1:3 ratio in serum-free Optimem and 1 hour incubation at room temperature (Millipore Sigma). The supernatant was replaced 4–6 hours post-transfection with 1 mL fresh medium. The supernatants were collected at 72 hours post-transfection and filtered using Pall 0.45 µm AcroPrep Advance Plate, Short Tip filter plates (polyethersulfone). For entry assays, luciferase measurements were conducted in the supernatant to normalize the amount of supernatant used for infection. The filtered supernatants were mixed with $2x10^4$ VAT or A549-ACE2$^h$ cells and plated in 96-well plates. Six hours post-infection, the cells were washed once with 300 µL PBS and 100 µL of fresh medium was added. Luciferase measurements in the supernatants were conducted at 24 hours post-infection using 50 µL of supernatant and 50 µL of nano-Glo luciferase reagent (Promega).

### Immunofluorescence microscopy

LD staining in cells infected with replicons was conducted as described previously [7]. For LD staining in NSP6-expressing cells, it was conducted as described previously [7] with minor modifications. Briefly, HuH-7.5 cells were transduced with lentiviral vectors expressing FLAG-tagged NSP6 mutants as described previously [7]. After selection, the cells were plated onto

poly-D-lysine treated 24-well glass-bottom plates (Corning) and 0.5 mM BSA-complexed oleic acid was added (Sigma Aldrich). The cells were incubated overnight, the culture medium was removed, cells were fixed with 4% paraformaldehyde, permeabilized with 0.1% TritonX-100 in PBS, and probed with monoclonal mouse-anti-FLAG M2 (Sigma, F1804-1MG, 1:500), rabbit-anti-ADFP (PLIN2, Invitrogen, PA5-29099, 1:200), goat-anti-rabbit-AlexaFluor546 (Invitrogen, A11010, 1:200), and donkey-anti-mouse-AlexaFluor488 (Invitrogen, A21202, 1:200). Cells were stained with LipidTox Deep Red (1:500) and Hoechst (1:500) in PBS, washed with PBS, and resuspended in PBS for imaging. Cells were imaged on an Olympus FV3000RS confocal microscope with a 60X objective, and LD fluorescence was quantified using the Imaris 9.9.1 software.

### Immunoprecipitation and Western blot analysis

HEK293T cells were co-transfected with NSP6 and mCherry-DFCP1 expression vectors using X-tremeGENE 9 DNA Transfection Reagent at a 1:3 ratio in serum-free Optimem (Millipore Sigma). After 72 hours, the cells were collected and protein isolated using IP lysis buffer (10 mM Tris HCl pH 7.5, 150 mM NaCl, 0.5 mM EDTA, 1% NP40) and quantified by DC Assay (BioRad). Two mg of protein was loaded onto 100 μl of M2 FLAG magnetic beads (Sigma) for 18–24 hours at 4 °C while rotating. Beads were washed 6 times, eluted with 3X FLAG peptide (Sigma) for 30 min at 4 °C shaking, 5 μg of input, and IP samples were loaded onto a Criterion acrylamide gel (BioRad) for Western Blot analysis. Antibodies used were Rabbit-anti-DFCP1 (Cell Signaling Technology, E9Q1S, 1:5000), monoclonal mouse-anti-FLAG M2 (Sigma, F1804-1MG, 1:5000), and monoclonal mouse-anti-GAPDH (ProteinTech, 60004-1-Ig, 1:5000).

### *In vitro* Infection experiments and plaque assay

VAT or A549-ACE2[h] cells were seeded into 12-well plates to reach ~70% confluency the next day. At the time of infection, viral inoculum was added to the cells at multiplicity of infection of 0.1. One to two hours after the addition of the inoculum, the cells were washed with PBS three times, and fresh medium was added. The supernatant and cells were harvested at 24, 48, and 72 hours post-infection for plaque assay and RT-qPCR, respectively, as described previously [8].

### *In vivo* infection and analysis

5 to 6-month-old heterozygous K18-hACE2 c57BL/6J mice were housed in groups of five per cage and provided a regular chow diet (The Jackson Laboratory, Bar Harbor, ME). Intranasal SARS CoV-2 BA.5 or BA.5 NSP6 F260L virus inoculation was conducted with $1 \times 10^5$ PFU per mouse under isoflurane anesthesia. Body weight of the mice was monitored and recorded daily post-infection to assess weight loss. Mice were euthanized at day 5 (n=10 per group; 5 male and 5 female) and day 14 (n=10 per group; 5 male and 5 female) post-infection, or earlier if they reached humane endpoint (≥20% weight loss), in accordance with established protocols to minimize distress. Organ samples from the lung, lymph node, and BALF were collected immediately after euthanasia. RNA extraction, RT-qPCR, plaque assay, flow cytometry, and histopathology were conducted as described previously [22].

### Supporting information

**S1 Fig. XBB-derived variants' mutations enhance viral RNA replication compared with the BA.2 variant. A and B)** Consensus ORF1ab mutations (>95% of sequences at time of emergence) between BA.2 and XBB (A) and between the XBB-derived variants (B). BA.2 fixed

mutations are indicated in red. **C and D)** Viral RNA replication was measured in infected VAT cells with indicated replicons and plotted as mean +/- SD of three independent biological replicates each conducted in triplicate. *, $p<0.05$; **, $p<0.01$; ***, $p<0.001$; ****, $p<0.0001$ by two-sided Student's T-test.
(TIF)

**S2 Fig. NSP6 L260F mutation enhances viral RNA replication and reduces LD content in infected cells.** Representative immunofluorescence images of VAT cells infected with indicated replicons and stained at 24 hours post-infection for LD and dsRNA. Quantification of these images is presented in Fig 3C.
(TIF)

**S3 Fig. NSP6 mutations do not modulate interactions with DFCP1.** Western blot analysis of FLAG-NSP6 immunoprecipitation from cells co-transfected with DFCP1 and indicated NSP6 mutants. The left half of the blot indicates immunoprecipitated samples and the right half is the corresponding input samples. The blot is representative of two independent biological replicates. The red arrows indicate observed bands for DFCP1.
(TIF)

**S4 Fig. BA.5 NSP6 F260L produces a slightly lower cellular immune responses than BA.5.** Flow cytometry analysis of indicated immune cell subsets in indicated organs was performed at the 5 days post-infection time-point of the *in vivo* experiment presented in Fig 4. The data is presented as mean +/- SD. *, $p<0.05$ by two-sided Student's T-test.
(TIF)

**S5 Fig. Structural analysis of NSP6 variants using AlphaFold. A)** The top 4 predicted structures of NSP6 WT (green), ΔSGF (blue), ΔSGF/L260F (red), and V24F/ΔSGF/R252K (yellow) were aligned using PyMOL. **B)** Close up of the second alpha helix on the luminal side of the ER. **C)** Close up of the cytoplasmic tail of NSP6 with the amino acids at positions 252 and 260 drawn as sticks.
(TIF)

**S1 Data. Source data file.** Excel file containing, in separate sheets, the underlying numerical data for Figs 2B, 2C, 3C, 3E, 3F, 3G, S2B, and S2C.
(XLSX)

AcknowledgmentThank you to Reena Zalpuri at the University of California Berkeley Electron Microscope Laboratory for advice and assistance in experiment planning, electron microscopy sample preparation, and data collection. Imaging and image processing was performed at the Gladstone Institutes' Assay Development and Drug Discovery Core and the Histology and Light Microscopy Core. We thank Veronica Fonseca and Alia Aguilar for administrative support.

## Author contributions

**Conceptualization:** Taha Y. Taha, Shahrzad Ezzatpour, Melanie Ott.

**Data curation:** Taha Y. Taha, Shahrzad Ezzatpour, Jennifer M. Hayashi, Chengjin Ye, Francisco J Zapatero-Belinchón, Julia A. Rosecrans, Gabriella R. Kimmerly, Irene P. Chen, Keith Walcott, Anna Kurianowicz, Danielle M. Jorgens, Natalie R. Chaplin, Annette Choi, David W. Buchholz, Julie Sahler, Zachary T Hilt, Brian Imbiakha, Cecilia Vagi-Szmola, Erica Stevenson, Martin Gordon.

**Formal analysis:** Taha Y. Taha, Shahrzad Ezzatpour, Jennifer M. Hayashi, Chengjin Ye, Danielle M. Jorgens.

**Funding acquisition:** Nevan J. Krogan, Melanie Ott.

**Investigation:** Taha Y. Taha, Shahrzad Ezzatpour, Jennifer M. Hayashi, Chengjin Ye, Francisco J Zapatero-Belinchón, Julia A. Rosecrans, Gabriella R. Kimmerly, Irene P. Chen, Keith Walcott, Anna Kurianowicz, Danielle M. Jorgens, Natalie R. Chaplin, Annette Choi, David W. Buchholz, Julie Sahler, Zachary T Hilt, Brian Imbiakha, Cecilia Vagi-Szmola, Erica Stevenson, Martin Gordon.

**Methodology:** Taha Y. Taha, Shahrzad Ezzatpour, Jennifer M. Hayashi, Chengjin Ye, Danielle M. Jorgens, Mauricio Montano.

**Project administration:** Taha Y. Taha.

**Resources:** Mauricio Montano, Danielle L. Swaney, Nevan J. Krogan, Gary R. Whittaker, Luis Martinez-Sobrido, Hector C Aguilar, Melanie Ott.

**Supervision:** Danielle L. Swaney, Nevan J. Krogan, Gary R. Whittaker, Luis Martinez-Sobrido, Hector C Aguilar, Melanie Ott.

**Validation:** Taha Y. Taha, Shahrzad Ezzatpour, Jennifer M. Hayashi, Chengjin Ye, Danielle M. Jorgens.

**Visualization:** Taha Y. Taha, Shahrzad Ezzatpour, Jennifer M. Hayashi, Chengjin Ye, Danielle M. Jorgens.

**Writing – original draft:** Taha Y. Taha.

**Writing – review & editing:** Taha Y. Taha, Shahrzad Ezzatpour, Jennifer M. Hayashi, Chengjin Ye, Francisco J Zapatero-Belinchón, Keith Walcott, Luis Martinez-Sobrido, Melanie Ott.

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
