## [Decision Letter · Decision Letter 0]

22 Jan 2025

PPATHOGENS-D-24-02742

Enhanced RNA replication and pathogenesis in recent SARS-CoV-2 variants harboring the L260F mutation in NSP6

PLOS Pathogens

Dear Dr. Ott,

Thank you for submitting your manuscript to PLOS Pathogens. After careful consideration, we feel that it has merit but does not fully meet PLOS Pathogens's publication criteria as it currently stands. Therefore, we invite you to submit a revised version of the manuscript that addresses the points raised during the review process.

Please submit your revised manuscript within 60 days Mar 23 2025 11:59PM. If you will need more time than this to complete your revisions, please reply to this message or contact the journal office at plospathogens@plos.org. Please include the following items when submitting your revised manuscript:

We look forward to receiving your revised manuscript.

Kind regards,

Tom Gallagher

Guest Editor

PLOS Pathogens

Michael Letko

Section Editor

PLOS Pathogens Sumita Bhaduri-McIntosh

Editor-in-Chief

PLOS Pathogens

orcid.org/0000-0003-2946-9497

Michael Malim

Editor-in-Chief

PLOS Pathogens

orcid.org/0000-0002-7699-2064

**Additional Editor Comments :**

Your submission was reviewed by three experts in the field. The reviewers noted the importance of the findings and were generally positive. However, reviewer 3 noted that proposed NSP6 operating mechanisms require additional experimental support. Specifically, the reviewer asked for greater evidence supporting the relationships between L260F substitution and LD, DMV, ER zippering, and DFCP1 interactions. These reasonable recommendations deserve attention during manuscript revision.

**Journal Requirements:**

1) Please upload all main figures as separate Figure files in .tif or .eps format. For more information about how to convert and format your figure files please see our guidelines: 

2) Some material included in your submission may be copyrighted. According to PLOSu2019s copyright policy, authors who use figures or other material (e.g., graphics, clipart, maps) from another author or copyright holder must demonstrate or obtain permission to publish this material under the Creative Commons Attribution 4.0 International (CC BY 4.0) License used by PLOS journals. Please closely review the details of PLOSu2019s copyright requirements here: PLOS Licenses and Copyright. If you need to request permissions from a copyright holder, you may use PLOS's Copyright Content Permission form.

Potential Copyright Issues:

i) Figures 2A, and 4A, 5D. Please confirm whether you drew the images / clip-art within the figure panels by hand. If you did not draw the images, please provide (a) a link to the source of the images or icons and their license / terms of use; or (b) written permission from the copyright holder to publish the images or icons under our CC BY 4.0 license. Alternatively, you may replace the images with open source alternatives. See these open source resources you may use to replace images / clip-art:

ii) The following Figure contains a logo or branding (Outbreak.info): S6. We are not permitted to publish this under our CC-BY 4.0 license, even with permission. We ask that you please remove or replace it.

3) We note that your Data Availability Statement is currently as follows: "All relevant data are within the manuscript and its Supporting Information files.". Please confirm at this time whether or not your submission contains all raw data required to replicate the results of your study. Authors must share the “minimal data set” for their submission. PLOS defines the minimal data set to consist of the data required to replicate all study findings reported in the article, as well as related metadata and methods (https://journals.plos.org/plosone/s/data-availability#loc-minimal-data-set-definition).

**Reviewers' Comments:**

Reviewer's Responses to Questions

**Part I - Summary**

Reviewer #1: This manuscript describes the role for a particular mutation in NSP6 L260F of SARS-CoV-2 on viral replication both in cultured cells and in a virus-susceptible mouse model. It was discovered that this mutation enhanced the replication of the viral RNA in both systems and this finding correlated with an enhanced host lipid droplet consumption by the mutated protein. In addition, this mutation also enhanced viral pathogenesis in mice compared to a F260L mutation in NSP6. This study provides novel and important insights by which non-spike proteins contribute to viral pathogenesis.

Reviewer #2: Taha and colleagues performed a comprehensive study on the evolution of SARS-CoV-2 variants, anlysinga mutations on the Spike and their impact on viral entry and replication. The authors also focused their analyses on NSP6, which is involved in viral RNA replication and lipid consumption. They report a negative correlation between viral entry and RNA replication: As variants evolved reduced entry functions due to immune pressure on Spike, RNA replication increased as a compensatory mechanism. They demonstrate that the frequent L260F mutation in NSP6 enhances viral infection in cells and increases pathogenesis in mice. The authors used a variety of experimental systems including novel tools, either Spike-defective or NSP6-defective genomes (replicons) of recent Omicron variants, which are clearly described and used. The study provides insights into evolutionary trajectories of recent variants and understanding of how the L260F mutation affects NSP6 function and viral replication. In summary, the manuscript demonstrates rigorous experimental techniques and provides novel insights into SARS-CoV-2 evolution.

Reviewer #3: The study investigates the role of NSP6 mutations, particularly L260F, in SARS-CoV-2 RNA replication and on host cell lipid droplets (LD). Using a replicon-based assay, the authors show enhanced replication in variants carrying the L260F mutation and explore its potential effects on LD dynamics. However, while the approach is valuable, the proposed mechanism through which NSP6 L260F would exert its role in viral replication remains unsupported by the data, as specified below.

**Part II – Major Issues: Key Experiments Required for Acceptance**

Reviewer #1: (No Response)

Reviewer #2: I don’t have major concerns.

Minor point

1. The authors could discuss the more recent evolution of the JN.1 lineage. The predominant lineages are KP.3.1.1 and XEC. How the spike mutations will impact viral entry. Do these strains carry additional mutations in NSP6?

Reviewer #3: Lipid droplets analysis

The authors report increased RNA replication in NSP6 L260F-containing variants such as BA.1, BA.2, and XBB.1.16 compared to wild-type NSP6. They show reduced mean fluorescence intensity (MFI) of LDs in these variants and conclude that enhanced LD consumption underlies the increased replication. However they do not mention other meaningful parameters, such as number and size of LDs. In addition, the reported differences in LD MFI are tiny and not fully consistent with the rate of RNA replication of the different virus variants. For instance, BA.1 exhibits the highest RNA replication, but its LD MFI value (400) is similar to that of BA.2 NSP6 L260F and XBB.1.16, which have one log lower replication rates. The lack of correlation between LD MFI and RNA replication undermines the proposed causative relationship. Furthermore, the differences in LD MFI are marginal (e.g., ~100 units) and accompanied by large data dispersion, raising questions about their biological relevance.

The authors use different clones stably expressing NSP6 mutants to analyze LDs, but the variability observed in LD measured parameters (also seen in the empty vector) makes it uncertain the interpretation of the reported tiny difference. Here the authors measure also the number and the size of LD: however no difference is measured in LD number and size between cells expressing wt NSP6 and cells transfected with the empty vector, reinforcing the suspect that the experimental system is not adequately set to assess LD dynamics. In addition the experimental design does not allow to discern whether the observed differences are due to altered LD biogenesis or LD consumption.

Additionally, NSP6 expression is not adequately controlled across the clones. In Fig. S4, large fields of NSP6-expressing clones show a low percentage of NSP6-positive cells, raising questions about whether mixed populations were used and the expression levels of NSP6 in these clones.

The authors fail to provide individual LD measurement values, but the graphs reveal substantial data dispersion (Fig. 3E-3F), while the differences among mutants are minimal (e.g., Fig. 3F). These small differences, even if statistically significant, seem unlikely to represent biologically meaningful changes. Further, in Fig. 3G, neither the graph’s scale nor the unit of measurement is provided, hindering interpretation of the data. These omissions and inconsistencies undermine the robustness of the authors’ conclusions regarding NSP6’s impact on LDs.

As in the case of cells infected with virus variants, also in the mutant NSP6 expressing cells only a correlative evidence is, at best, presented and no causative link is either explored or demonstrated.

ER Zippering and DMVs morphology

The authors conclude that L260F neither affects the ER zippering activity of NSP6 nor impacts the DMV biogenesis, but the data are insufficient to support this conclusion. The EM images in Figure 4b are low-resolution and fail to clearly depict DMVs or zippered ER membranes. Quantitative analysis of these structures is absent, making it impossible to draw meaningful comparisons. Higher resolution EM images and a more comprehensive analysis of DMV morphology are needed.

DFCP1-NSP6 interaction

In the co-IP experiments examining DFCP1-NSP6 interactions, the western blots show different molecular weights for DFCP1 in the input compared to the IP samples, as well as multiple bands for NSP6 that do not correspond to its expected ~35 kDa size. These anomalies are not explained, raising concerns about the identity of the observed bands. Hence, the authors claim no differences in DFCP1 interactions between NSP6 mutants and wild-type, but the data presented are insufficient to draw this conclusion.

In conclusion, while the study provides compelling evidence that NSP6 mutations affect viral replication, the evidence is not robust enough to support the proposed mechanisms. The observed differences in LD MFI are minor and inconsistent, and the link between LD consumption and replication is not established. The study of DMVs, ER zippering, and DFCP1 interactions requires better experimental controls, more detailed analyses, and higher-resolution imaging.

**Part III – Minor Issues: Editorial and Data Presentation Modifications**

Reviewer #1: Comments:

1. Both NSP1 and NSP6 affect viral RNA abundances. Explain for the more general reader the mechanistic differences by which these two proteins modulate viral gene expression.

2 Fig. 2B,C. XBB seems to display enhanced RNA replication., Can you comment on its genotype in relation to XBB.1.6 and BQ.1.1?

3. Fig. 3D. Panels 2-4 are difficult to read. Provide a higher resolution figure.

4. The phenotype of NSP6 L60F versus NSP6 F260L is striking. Can the authors provide a molecular docking model which may give some clues?

Reviewer #2: (No Response)

Reviewer #3: (No Response)

PLOS authors have the option to publish the peer review history of their article (what does this mean?). If published, this will include your full peer review and any attached files.

Reviewer #1: No

Reviewer #2: No

Reviewer #3: **Yes: **Antonella De Matteis

**Figure resubmission:**
---

## [Editor Report · Decision Letter 1]

3 Mar 2025

Dear Dr. Ott,

We are pleased to inform you that your manuscript 'Enhanced RNA replication and pathogenesis in recent SARS-CoV-2 variants harboring the L260F mutation in NSP6' has been provisionally accepted for publication in PLOS Pathogens.

Best regards,

Tom Gallagher

Guest Editor

PLOS Pathogens

Michael Letko

Section Editor

PLOS Pathogens

Sumita Bhaduri-McIntosh

Editor-in-Chief

PLOS Pathogens

orcid.org/0000-0003-2946-9497

Michael Malim

Editor-in-Chief

PLOS Pathogens

orcid.org/0000-0002-7699-2064
---

## [Editor Report · Acceptance letter]

Dear Dr. Ott,

We are delighted to inform you that your manuscript, "Enhanced RNA replication and pathogenesis in recent SARS-CoV-2 variants harboring the L260F mutation in NSP6," has been formally accepted for publication in PLOS Pathogens.

Best regards,

Sumita Bhaduri-McIntosh

Editor-in-Chief

PLOS Pathogens

orcid.org/0000-0003-2946-9497

Michael Malim

Editor-in-Chief

PLOS Pathogens

orcid.org/0000-0002-7699-2064